# Maternal Obesity in Twin Pregnancy: The Role of Nutrition to Reduce Maternal and Fetal Complications

**DOI:** 10.3390/nu14071326

**Published:** 2022-03-22

**Authors:** María de la Calle, Jose L. Bartha, Clara Marín, Juan Carlos Rus, Guillermo Córcoles, Santiago Ruvira, David Ramiro-Cortijo

**Affiliations:** 1Obstetrics and Gynecology Service, Hospital Universitario La Paz, Paseo de la Castellana 261, 28046 Madrid, Spain; maria.delacalle@uam.es (M.d.l.C.); joseluisbartha@me.com (J.L.B.); cmbarbancho@gmail.com (C.M.); juancarlos_rs20@hotmail.com (J.C.R.); guillecorcor@gmail.com (G.C.); 2Department of Physiology, Faculty of Medicine, Universidad Autónoma de Madrid, C/Arzobispo Morcillo 2, 28049 Madrid, Spain; santiago.ruvira@estudiante.uam.es

**Keywords:** maternal obesity, body mass index, twin pregnancy, nutritional control, high-risk pregnancy, obstetric adverse outcomes

## Abstract

There are more and more obese mothers with twin gestations. For a long time before, the responses of lymphocytes and platelets in obese women can cause a low-grade inflammation. In addition, a proper control of gestational weight gain would improve the outcomes in mothers with high pre-gestational body mass index (BMI). In women with high pre-gestational BMI and twin pregnancy, our aims were to explore the biochemical and hematological parameters and to study the rate of obstetric adverse outcomes. This was an observational and retrospective study conducted in the Hospital Universitario La Paz (Madrid, Spain). We included 20 twin pregnancies as the lean group (BMI = 18.5–24.9 kg/m^2^), homogeneous in the maternal age and ethnicity, and having parity with other 20 twin pregnancies as the obese group (BMI ≥ 30 kg/m^2^). The maternal data and maternal, fetal, obstetric, and neonatal complications were collected from the medical records. In the first and third trimester of pregnancy, the biochemical and hematological parameters of the blood were assayed. In this cohort, gestational weight gain was significantly lower in the obese than lean group. In the first trimester, the hemoglobin levels in obese women (12.1 ± 0.8 g/dL) were lower than lean women (12.6 ± 0.7 g/dL; *p*-Value = 0.048). In addition, the tendency of glucose levels, TSH levels and platelets was to increase in obese compared to lean women. In the third trimester, the TSH levels were higher in obese (3.30 ± 1.60 mUI/L) than lean women (1.70 ± 1.00 mUI/L; *p*-Value = 0.009). Furthermore, there was a tendency for levels of platelets and lymphocytes to increase in obese compared to lean women. No significant differences were detected in the rate of maternal, fetal, obstetrical, and neonatal complications between the groups. The hemoglobin, platelets, lymphocytes and TSH levels need further investigation to understand potential subclinical inflammation in obese women. Furthermore, obese women with twin pregnancies should follow-up with a specialist nutritionist, to help them control their gestational weight gain with appropriate dietary measures.

## 1. Introduction

The high prevalence of obesity makes it more likely that more women in reproductive age with a high body mass index (BMI) will become pregnant [1]. The Spanish Institute of Statistics reflects that the percentage of obesity (BMI ≥ 30 kg/m^2^) in women aged 18–44 years is approximately 10.5%, considering overweight women (BMI = 25–29.9 kg/m^2^) the prevalence increases to 20.4% [2]. Maternal obesity has been shown to increase short- and long-term maternal, obstetric, and neonatal complications [3]. These women are at increased risk of developing gestational diabetes, pregnancy-induced hypertension, preeclampsia, prolonged labor, shoulder dystocia, prematurity, and C-section, as well as fetal malformations, fetal growing disbalance, and increased perinatal mortality [4].

Furthermore, the relationship of chronic inflammation and obesity has been extensively reported. This systemic inflammation can be associated with obstetric adverse outcomes such as gestational diabetes and preeclampsia [5,6,7]. However, when these clinical manifestations are shown, the chronic inflammation could have been running for much longer. The responses of leukocytes cells, particularly lymphocytes, induced by adipocyte hypertrophy in obese women can cause a potential low-grade inflammation [8]. In this context, lymphocytes and platelets, among other biomarkers, have been proposed as potential diagnostic tools for this subclinical inflammatory status in obese pregnant women [9]. Dietary pattern can have a significant effect on adipose tissue and fatty acids composition [10], which can modify the inflammation process [10,11]. There is evidence that diets high in animal protein and low in fruits and vegetables may contribute to an increase in inflammatory markers during pregnancy [12]. For this reason, nutritional follow-up of pregnancies in obese women becomes extremely important.

The interaction between gestation, obesity and inflammation may have implications for the mother, the fetus, and the newborn. Pre-gestational obesity and also excessive gestational weight gain during pregnancy could lead to metabolic syndrome and cardiovascular disease in the infants. There are also long-term risks for descendants, such as obesity and increased early adult morbidity and mortality [13]. Although high pre-gestational BMI is associated with adverse obstetric and fetal outcomes, proper gestational weight gain control would improve the outcomes [14,15]. In this context, the nutritional intervention and follow-up of the high-risk pregnancy can be crucial to prevent complications considering the identified predictors. In pregnant women, it would be necessary to discuss the nutritional counseling and dietary advice in antenatal care with expert nutritionists, which has been effective in preventing other complications, such as iron deficiency and anemia [16].

Twin pregnancies are associated with maternal and fetal adverse outcomes. The maternal outcomes could be led by hemodynamic changes that are more pronounced than in singleton pregnancies. Maternal complications such as pregnancy-induced hypertension, preeclampsia, gestational diabetes, intrahepatic cholestasis, or iron deficiency anemia, among others, are more frequent in twin than single pregnancies [17]. Fetal complications include an increased risk of intrauterine growth restriction, congenital anomalies, and prematurity [17]. However, twin pregnancy usually has been excluded from obstetrical studies as it is considered to be a confusion factor. Conversely, the increase in the age of women using assisted reproduction techniques to access to maternity [17,18] is changing the maternity profile in developed countries, leading to an increase in twin pregnancies. In addition, obesity is one of the causes of sterility [5], and women could be using assisted reproductive technologies to achieve pregnancy.

The rise in obesity coupled with the delay in maternity have increased the incidence of multiple gestations in obese women. However, there are hardly any studies on this new profile of obese women with twin pregnancies, which is clearly increasing in our society.

Considering the above, we hypothesize that close nutritional follow-up in obese women with twin pregnancies can prevent obstetric adverse outcomes. However, the high pre-gestational BMI may represent a low-grade inflammation that may alter hematologic markers throughout pregnancy. In a selective cohort of twin-pregnant women with high pre-gestational BMI and close control and follow-up by nutritionists’ team, our aims were to explore the biochemical and hematological parameters along gestation, detecting potential alterations, and to study the rate of maternal, fetal, obstetrical, and neonatal outcomes, thus discovering if nutritional control would prevent these adverse obstetrical outcomes.

## 2. Materials and Methods

### 2.1. Study Design and Cohort Selection

This was an observational and retrospective study conducted in the high-risk unit at Obstetric Service of the Hospital Universitario La Paz (HULP, Madrid, Spain); this unit is supported by a multidisciplinary team including nutritionists. Data was collected from the medical records of twin gestations delivered between 1 January and 31 December of 2019. During this period, 223 twin pregnancies were reviewed. If pregnancy was diagnosed with pre-gestational risk factors (such as hypertension, diabetes mellitus, or chronic liver disease), these were excluded to avoid potential bias in the variables of the study. Twin pregnancies with first-trimester abortions (non-progress of gestation) were not collected. Additionally, the body mass index (BMI, kg/m^2^) prior to being pregnant was considered as an inclusion criterion. These data were available when pregnancy was followed-up in HULP, but if BMI was not available then the medical record was also discarded. BMI ≥ 30 kg/m^2^ was studied as obesity group. In total, 20 pregnancies were enrolled in obese group (Figure 1). The obese group was closely monitored by the nutritionists, who scheduled an appointment every 15 days to control their weight, diet, as well as the physical activity (to walk at least 30–40 min every day). The nutritional advice was given from the beginning of pregnancy: a balanced 2500 Kcal diet with an adequate intake of nutrients, both in the amount and type of macronutrients (proteins, fats, and carbohydrates). In the healthy group, we included proteins which provided 10–35% of energy, fats 20–35% of energy and carbohydrates 45–65% of energy. In our country, the Mediterranean diet provides a sufficient quantity of immediate principles that guarantees the health of both the mother and the fetus.

The comparison group was considered a pre-gestational BMI in the range 18.5–24.9 kg/m^2^ (lean group). We included 20 pregnancies in lean group, homogeneous in the maternal age and ethnicity, and having parity to avoid potential confusion factors (Figure 1).

### 2.2. Variable Collection

The variables extracted from the medical records were:

Maternal data: age (years), pre-gestational weight (kg) and BMI (kg/m^2^), body weight gain during pregnancy (kg), ethnicity (categories: Caucasian, Latin and Black), smoking habits (yes/no), parity, miscarriages, assisted reproduction techniques (yes/no) and twin chorionicity extracted by echography as dichorionic diamniotic or monochorionic diamniotic.

**Maternal biochemical and hematological variables**: blood sample was extracted from the women by using venipuncture in Vacutainer^®^ tubes, in fasting state, from 8:00 to 9:00 a.m., following the protocols established by the medical staff. The plasma was obtained by centrifugation (2100 rpm, 15 min at 4 °C), within a maximum of 2 h after extraction. The biochemical and hematological parameters were assayed by Laboratory Medicine Service from HULP according to hospital guidelines. The variables analyzed were glucose (mg/dL), hemoglobin (g/dL), fibrinogen (mg/dL), ferritin (g/mL), TSH (mUI/L), vitamin D (ng/mL), hematocrit (%), platelets (10^6^/mL), leukocytes (10^6^/mL), lymphocytes (10^3^/μL), neutrophils (10^3^/μL), monocytes (10^3^/μL), eosinophils (10^3^/μL), basophils (10^3^/μL). These variables were collected at first (11th ± 1 weeks) and third (35th ± 1 weeks) trimester of pregnancy, to test the alterations in these biochemical variables at the beginning and at the end of pregnancy, being two key points in the follow-up.

The follow clinical variables were collected of the medical records if the obstetrical complications were diagnosed according to the medical staff and considering the protocols of the Spanish Society of Gynecology and Obstetrics (SEGO guidelines).

**Maternal complications**: these variables were pregnancy-induced hypertension (defined as systolic blood pressure ≥ 140 mmHg and/or diastolic ≥ 90 mmHg blood without proteinuria after 20 weeks of gestation), preeclampsia (defined as pregnancy-induced hypertension and at least one of the following criteria: proteinuria (protein/creatinine ratio = 30 mg/mol or 24 h urine protein 300 mg), clinical organ dysfunction, or utero-placental dysfunction by echo-Doppler), gestational diabetes mellitus (defined with oral glucose overload test of 100 g after a previously altered oral glucose overload test of 50 g), iron deficiency anemia (defined as a hemoglobin < 11.0 g/dL in the first and third trimester and <10.5 g/dL in the second trimester), intrahepatic cholestasis (defined as palmoplantar pruritus during the third trimester associated with alterations in liver enzymes or increased total bile acids), urinary tract infection (defined as morphological and/or functional alterations with >100 × 10^3^ colony-forming units in urine culture) and maternal death.

**Fetal complications**: these variables were intrauterine growth restriction (defined as the fetal weight < 3rd percentile or <10th percentile with altered feto-placental by echo-Doppler), fetal malformations assayed by echography principally in cardiac and nervous system and genitor-urine tract, and fetal death.

**Obstetrical complications**: these variables were premature rupture of membrane (defined as rupture of the amniotic membranes occurring before the onset of spontaneous labor), threat of premature labor (characterized by regular uterine contractions with cervical modifications occurring between 22 and 36 weeks of gestation with integral amniotic membranes), route of delivery (vaginal/C-section), and puerperal hemorrhage (vaginal bleeding of >0.5 L after vaginal or >1 L after C-section, or bleeding that cause hemodynamic instability of the women).

**Neonatal variables and complications**: the variables considered were the gestational age at delivery (weeks), prematurity (gestational age < 37 weeks), birth body weight (grams), low birth weight (LBW, birth body weight < 2500 g), Apgar score at 5 min, artery pH, Neonatal Intensive Care Unit admission (NICU, yes/no), respiratory distress syndrome (RDS, yes/no), and neonatal death.

### 2.3. Statistical Analysis

In quantitative analysis, the variable distribution was tested by Shapiro test. Based on variable distribution, these variables were expressed as median and interquartile range [Q1; Q3] or median and standard error of mean (SEM). In addition, the Mann–Whitney and Student’s *t* tests were used to compared quantitative variables. In biochemical and hematological variables, which were collected longitudinally, paired test comparison between groups was applied. Qualitative variables were summarized as sample size and relative frequency (%) and Fisher exact test was used to compared contingency tables. The *p*-Value < 0.05 were considered statistically significant.

Statistical analysis was performed with R software (version 4.1.2, 2021, R Core Team, Vienna, Austria) within RStudio interface using *rio*, *tidyverse*, and *dplyr* packages.

## 3. Results

### 3.1. Maternal Characteristics of the Cohort

In the cohort, no significant differences were detected in maternal age, ethnicity, smoking habits, parity, miscarriages, and twin chorionicity between groups. As expected, weight before pregnancy and BMI were significantly higher in obesity than lean women, and body weight gain during pregnancy was significantly lower in obesity than lean women. In addition, there was a higher rate of assisted reproduction techniques in lean than obese women (Table 1).

### 3.2. Biochemical and Hematological Parameters

At the beginning of pregnancy, the hemoglobin levels in obese women were significantly lower than lean women (Table 2). In addition, while there were no significant differences between groups, the vitamin D levels were also lower in obese than lean women. Contrary, the glucose and TSH levels and platelets were higher in obese compared to lean women.

At the end of pregnancy, the TSH levels were significantly higher in obese than lean women (Table 2). Furthermore, close to significantly, the levels of platelets and lymphocytes were higher in obese compared to lean women. In addition, TSH levels had a significantly pronounced increase from the beginning to the end of pregnancy in obese compared to lean women.

### 3.3. Clinical Variables

In this cohort, there was no maternal death either during gestation or at labor. There was also no intrahepatic cholestasis. No significant differences were detected in the ratios of maternal, fetal, obstetrical, and neonatal complications between the groups (Table 3). Although there were no significant differences, the obese women had lower gestational age at delivery compared to the lean women (lean = 37.5 [36.5; 38.0] weeks, obese = 37.1 [35.0; 37.4] weeks; *p*-Value = 0.083). On the other hand, the only neonatal death was found in the obese group.

## 4. Discussion

In this study, we explore twin pregnancies, a type of gestation that is frequently excluded from studies in the obstetric field, but whose incidence is rising in developed countries. Additionally, we have studied obese women, as a high-risk pregnancy, to highlight the importance of following up closely under a multidisciplinary team that included nutritionists. Therefore, not only twin pregnancies but also obesity is becoming more frequent in developed countries. For this reason, it seemed highly relevant to evaluate obstetric complications in this emerging group of pregnant women. Recommendations on weight normalization before becoming pregnant or undergoing assisted reproduction techniques should be made to reduce the risk of developing complications during pregnancy. This context is of special interest in the obstetric field, as mentioned in a review by Prof. Judith Stephenson which examined the evidence of the importance of nutrition in the preconception period [19].

It has been shown that obesity can increase the risk of miscarriage and negatively influences the success of assisted reproduction techniques. The mechanisms underlying the effect of obesity on the outcome of these techniques could be explained by alterations in the hypothalamic–pituitary–ovarian axis. In addition, it also appears that obesity may negatively affect oocyte quantity and quality and alters endometrial decidualization [20]. In our study, obese women had lower assisted reproduction techniques compared to lean women. This finding could probably be due to the high rate of unsuccessful in vitro fertilization in obese women [21]. In Spain, it is advisable to have a normal BMI before starting reproductive treatment to guarantee its success, which is the reason why we found more fertilization techniques in the group of lean women. Considering our study design, we did not have a miscarriage rate since we only studied successful pregnancies with live newborns above 36 weeks.

There is evidence related to the relationship between anemia and obesity due to an inadequate diet that does not cover iron requirements during pregnancy [22]. Additionally, obese women are known to have increased hepcidin concentrations (presumably by chronic inflammation) and, therefore, reduced iron absorption and decreased hemoglobin levels [23]. It is interesting to note that these studies were reported in single pregnancies, but twin pregnancy could be an additional risk factor in the development of anemia. It has been demonstrated that anemia is more frequent in twin pregnancies, since they are accompanied by greater hemodynamic changes and requirement of maternal blood for the feto-placental growth, since there are two fetuses [17]. According to our data, we detected low hemoglobin levels in the first trimester in obese pregnant women. However, we did not find a relationship between anemia and obesity. In fact, we observed that the incidence of iron deficiency anemia was less in obese pregnant women. According to HULP guidelines, twin pregnancies are followed up closely by four hemograms and ferritin determination during the gestation, which allows for an early treatment of anemia and iron deficiency. Importantly in HULP, all pregnant women were receiving iron supplements and recommendations on diets rich in iron from the beginning of pregnancy. According to our data, vitamin D levels also tend to be low at the beginning of pregnancy in obese compared to lean women. These results demonstrate the importance of requesting not only hemoglobin levels (performed in almost all countries) but also vitamin D levels in obese pregnant women, since we demonstrated a deficiency that can be corrected from the first trimester with the administration of this vitamin. Hypovitaminosis D in pregnancy contributes to the risk of adverse obstetrical outcomes including preeclampsia, fetal growth restriction and preterm birth [24]. Early detection in pregnancy, particularly in obese pregnant women, will allow us to administer vitamin D supplements and would lead to a reduction in the aforementioned complications. In HULP and in their primary care center, all pregnant women received nutritional education and were informed of the importance of exposure to the sun one hour a day in contributing to an increase in vitamin D levels.

Related to inflammation, women with high gestational weight gain expressed higher levels of platelets, suggesting a subclinical inflammation associated with excessive fat accumulation [9]. Furthermore, it was hypothesized that lymphocyte is a better indicator of nutritional status and general stress in obesity [25]. We found less gestational weight gain in obese group compared to lean pregnant women; it is likely that obese women followed the nutritionist specialist team recommendations for optimal body weight gain during gestation. Surprisingly, we continued to observe a high tendency level of platelets in first and third trimesters and also high trends of lymphocytes levels in the third trimester. These data could suggest a possible metabolic misbalance in a subclinical inflammation context. Notably, it is described that the inflammation can transfer from mother to fetus [9] and lead to an unhealthy process.

The control of gestational weight gain in twin pregnancies is highly important not only for its subclinical but also for its clinical implications. Several studies described that excessive gestational weight gain is associated with obstetrical adverse outcomes. Women who did not meet the guidelines for gestational weight gain had a high rate of fetal growth disbalance, premature rupture of membranes, C-section, and prematurity [3,15]. Adherence to a healthy diet during pregnancy, such as a Mediterranean diet, prevented excessive gestational weight gain and postpartum weight retention [26,27]. In addition, a meta-analysis shows that a balanced diet reduces the risk of preeclampsia, gestational diabetes, pregnancy-induced hypertension and preterm delivery [14], particularly important in women with high pre-gestational BMI. In this sense, our results in gestational weight gain of obese women, controlled by nutritionists, could have contributed to better knowledge in obstetrical outcomes. Additionally, we encourage our pregnant women to walk every day for at least one hour, especially the obese women. The twin-pregnant women could be more body movement limited, thus the best exercise would be walking and swimming, considering the high risk of prematurity due to abdominal distension.

Related to maternal complications, other studies have observed a negative influence of high pre-gestational BMI on maternal complications [28,29]. Single pregnancies with high pre-gestational BMI have been reported as having an increased risk of pregnancy-induced hypertension and gestational diabetes [1]. However, little data has been explored in twin pregnancies. We have observed that women with pre-gestational obesity and twin gestations also have an increased rate of pregnancy-induced hypertension and gestational diabetes. Therefore, lifestyle interventions, including dietary and physical activities, are effective first-line strategies for gestational diabetes prevention, especially in the high-risk group, such as obese pregnant women with twins, where it should be recommended [30]. On the other hand, there are studies that analyze the risk of preeclampsia in twin pregnancies. There is evidence of an increased risk of preeclampsia in overweight and obese mothers [28,29]. However, although in single pregnancies the pre-gestational BMI could be useful to predict preeclampsia [31], in twin pregnancies, this indicator might not be a useful parameter to predict preeclampsia [32]. In our study, we did not observe a higher incidence of preeclampsia in the obese women with twin pregnancies. This finding may be explained by the fact that, in HULP, all obese women with twin gestations (and other added risk factor for preeclampsia) were receiving 150 mg of acetylsalicylic acid from week 12 of gestation, a measure that has been shown to prevent this complication [33].

Related to obstetric complications, a systematic review has shown that maternal obesity constitutes a risk factor for preterm delivery in twin pregnancies [34]. However, the higher rate of prematurity in pregnancies of obese women published could be due to the higher programed deliveries for related maternal disorders, such as pregnancy-induced hypertension, gestational diabetes, and preeclampsia [35]. Fortunately, our study in twin pregnancies, we did not observe differences in the prematurity in relation to pre-gestational BMI. Although there was more gestational diabetes and pregnancy-induced hypertension in obese twin gestations, these complications were not a reason for prematurity. It would be pertinent to highlight the closely followed-up gestational control in specialized units to prevent these obstetric adverse outcomes.

Additionally, the frequency of preterm delivery due to premature rupture of membranes, cervical shortening, or uterine dynamics, is increased in obese women with twin pregnancies [36]. We also found an increasing trend of premature rupture of membranes in obese women with twin gestation compared to lean women. However, this did not lead to an increase in prematurity, potentially because in HULP, twin-pregnant women are monitored closely up to the 34th week of gestation and, in the case of premature rupture of membranes, pregnant women are hospitalized with antibiotics, resting and tocolytic drugs to prevent chorioamnionitis and prematurity. Likewise, the exhaustive control of the cervix from the 20th week of gestation and the insertion of cervical pessary in a short cervix have allowed us to reduce prematurity independently of the BMI of the woman.

Obese women are likely to have a failed induction of labor, non-progressive labor dystocia, meconium amniotic fluid, and fetal malpresentation [37]. Factors that lead the C-section procedures. It has been described an increase in the C-section in single pregnancies when the BMI of women was increased [1,37]. In our cohort of twin pregnancies, the C-section was more frequent in the group of obese compared to lean women. This finding agrees with the Al-Obaidly et al. data [28].

Regarding fetal and neonatal outcomes, evidence suggests that twin gestations are associated with a high rate of intrauterine growth restriction, congenital anomalies, and fetal death. In addition, there is an increased risk of adverse neonatal outcome [17]. In contrast, neonates of obese women are more likely to be large for gestational age, having a higher birthweight [38]. This increase in gestational age and birthweight could be led by the thyroid function abnormalities in obese women. The BMI is positively correlated with TSH levels [39,40]. Previous data reveal that the maternal thyroid levels during pregnancy may influence childhood adiposity and cardiovascular development. Lower maternal TSH level was associated with lower childhood BMI [41]. Furthermore, maternal thyroid hormones levels during pregnancy have been implicated in the etiology of obesity in descendants [42], although their role remains unknown. However, the largest body of evidence has been reported in women with single pregnancies. Interestingly, our data on obese women with twin pregnancy shows high levels of TSH in the first and third trimester, even when compared longitudinally. However, and fortunately, no differences in fetal growth and malformations were detected. This is probably due to the close follow-up of these high-risk pregnancies by the nutritionist team. It is interesting to study the maternal thyroid function, their relationship with maternal nutrition and BMI, and the impact on fetal outcome, particularly important in twin pregnancy, which usually is excluded in obstetric studies. 

Finally, a meta-analysis indicates that maternal obesity is associated with an increased risk of fetal death, although the mechanisms to explain this association are not clear [43]. In our study, we did not observe more fetal malformations or stillbirth in obese twin pregnancies, due to the close control of the fetuses using Doppler ultrasound and fetal heart monitoring. We did not observe a higher frequency of admission to the NICU between groups. These findings are consistent with Al-Obaidly et al., who observed no differences between lean and obese groups in relation to NICU admission in twin pregnancies [28]. Similarly, Balki et al. also found no relationship between maternal BMI and length of hospital stay in twin pregnancies [44].

### Strengths and Limitations

The main limitations of our study derive from its retrospective design and the small sample size. There are no major differences between groups, although studies in other contexts did not find them either [15]. Women with high pre-pregnancy BMI in twin gestations who gain the optional amount of weight in pregnancy improve pregnancy outcomes. In our study, the obese women are closely monitored by the nutrition unit. In addition, our results may be useful because they derive from a Spanish population, representative of pregnant women with twin pregnancies, and followed up in a referral hospital. Unlike previous studies that focus on a single aspect of pregnancy, we have analyzed maternal, obstetric, and fetal complications as well as hematological and biochemical analysis. However, it should be considered that in mid-pregnancy, hematological and biochemical parameters may undergo variations that should be explored.

## 5. Conclusions

Conclusively, in women with increased pre-gestational BMI and twin pregnancies, we observed a high tendency of gestational diabetes and pregnancy-induced hypertension. Therefore, lifestyle interventions including dietary and physical activities should be highly recommended to prevent these obstetrical complications. In addition, vitamin D levels should be measured in all obese pregnant women with twins in the first trimester, since a decrease in those levels has been shown. Vitamin D supplementation from the beginning of the pregnancy could decrease adverse obstetrical outcomes. Additionally, we found iron deficiency in obese women with twin pregnancies. Hence, iron supplements and recommendations on iron-rich diets from the beginning of pregnancy could reduce anemia in pregnant women. However, the hemoglobin, platelets, lymphocytes and TSH levels need further investigation to understand if obese women have subclinical inflammation that could compromise the obstetric outcomes. Obese women with twin pregnancies should be followed closely by a multidisciplinary team including midwives and a specialist nutrition unit, to detect early obstetrical complications and to control their gestational weight gain with appropriate dietary measures and lifestyle recommendations.

## Figures and Tables

**Figure 1 nutrients-14-01326-f001:**
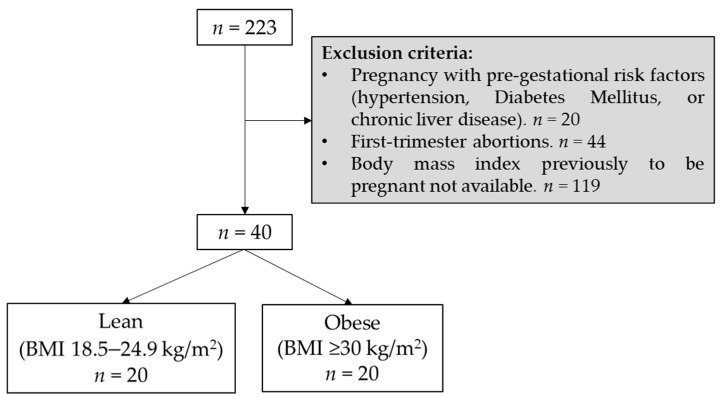
The confidentiality and anonymity of the data was guaranteed in every moment of the study protocol.

**Table 1 nutrients-14-01326-t001:** Maternal characteristics between groups.

	Lean (*n* = 20)	Obesity (*n* = 20)	*p*-Value
Maternal age (years)	37.0 [35.8; 43.0]	36.0 [32.8; 38.5]	0.097 ^a^
BMI (kg/m^2^)	21.4 [20.3; 22.8]	32.9 [31.2; 34.5]	<0.001 ^a^
Ethnicity			0.999 ^b^
Caucasian	4 (20.0%)	4 (20.0%)	
Latin	16 (80.0%)	15 (75.0%)	
Black	0 (0.0%)	1 (5.0%)	
Smoking habits	2 (10.0%)	1 (5.0%)	0.999 ^b^
Parity	1.5 [1.0; 2.0]	2.0 [1.0; 3.2]	0.173 ^a^
Miscarriages	0.0 [0.0; 1.0]	0.0 [0.0; 1.0]	0.659 ^a^
Weight pre-pregnancy (kg)	58.0 [55.0; 63.5]	89.7 [80.8; 94.5]	<0.001 ^a^
Weight gain during pregnancy (kg)	14.8 (5.9)	8.8 (6.1)	0.003 ^c^
Twin			0.273 ^b^
Dichorionic diamniotic	13 (65.0%)	17 (85.0%)	
Monochorionic diamniotic	7 (35.0%)	3 (15.0%)	
Assisted reproduction techniques	14 (70.0%)	3 (15.0%)	0.001 ^b^

In non-normal variables, the data show median and interquartile range [Q1; Q3], in normal variables, the data show mean and standard error of mean (SEM). In qualitative variables, the sample size (*n*) and relative frequency (%) are shown. The *p*-Value was extracted by ^a^ Mann–Whitney, ^b^ Fisher’s and ^c^ Student’s *t* tests. Body mass index (BMI).

**Table 2 nutrients-14-01326-t002:** Biochemical and hematological parameters at the beginning and the end of pregnancy between groups.

	Trimester 1			Trimester 3			*p*-Value ^3^
	Lean (*n* = 17)	Obesity (*n* = 15)	*p*-Value ^1^	Lean (*n* = 20)	Obesity (*n* = 19)	*p*-Value ^2^	
Glucose (mg/dL)	78.2 (7.50)	82.7 (6.60)	0.096 ^a^	75.7 (7.50)	75.9 (7.90)	0.932 ^a^	0.438
Hemoglobin (g/dL)	12.6 (0.70)	12.1 (0.80)	0.048 ^a^	12.3 (1.00)	12.3 (0.80)	0.858 ^a^	0.224
Fibrinogen (mg/dL)	494.1 (101.8)	592.5 (182.2)	0.373 ^a^	604.5 [580.8; 650.8]	651.0 [590.0; 708.0]	0.190 ^b^	0.905
Ferritin (g/mL)	22.0 [14.5; 31.5]	15.5 [10.0; 37.2]	0.565 ^b^	24.0 [15.5; 32.0]	19.0 [16.0; 29.0]	0.884 ^b^	0.145
TSH (mUI/L)	1.30 (1.10)	2.20 (1.30)	0.051 ^a^	1.70 (1.00)	3.30 (1.60)	0.009 ^a^	0.048
Vitamin D (ng/mL)	20.7 (8.40)	13.0 (6.40)	0.070 ^a^	21.6 (5.70)	17.4 (7.30)	0.323 ^a^	0.675
Hematocrit (%)	38.0 (2.1)	37.4 (2.10)	0.469 ^a^	37.4 [35.8; 38.7]	37.1 [35.5; 39.0]	0.725 ^b^	0.919
Platelets (10^6^/mL)	245.0 [201.5; 267.5]	289.0 [241.0; 331.0]	0.056 ^b^	206.5 [188.5; 250.5]	253.0 [204.0; 309.0]	0.074 ^b^	0.912
Leukocytes (10^6^/mL)	8.90 (1.90)	8.80 (2.20)	0.892 ^a^	8.90 [8.10; 9.10]	8.50 [7.80; 1.10]	0.955 ^b^	0.703
Lymphocytes (10^3^/μL)	2.13 (0.82)	2.22 (0.66)	0.737 ^a^	1.96 (0.59)	2.28 (0.57)	0.091 ^a^	0.588
Neutrophils (10^3^/μL)	6.00 (1.42)	5.66 (1.61)	0.546 ^a^	5.87 (1.04)	6.04 (1.43)	0.670 ^a^	0.689
Monocytes (10^3^/μL)	0.49 (0.16)	0.48 (0.13)	0.860 ^a^	0.49 [0.43; 0.60]	0.44 [0.38; 0.60]	0.800 ^b^	0.515
Eosinophils (10^3^/μL)	0.08 [0.06; 0.20]	0.12 [0.09; 0.24]	0.183 ^b^	0.10 [0.07; 0.14]	0.10 [0.07; 0.21]	0.473 ^b^	0.943
Basophils (10^3^/μL)	0.03 [0.02; 0.04]	0.03 [0.03; 0.05]	0.410 ^b^	0.03 [0.02; 0.04]	0.03 [0.02; 0.05]	0.689 ^b^	0.555

In non-normal variables, the data show median and interquartile range [Q1; Q3], in normal variables, the data show mean and standard error of mean (SEM). The *p*-Value ^1^ and ^2^ compared group within the trimester and were extracted by ^a^ Mann–Whitney and ^b^ Student’s *t* tests. The *p*-Value ^3^ was extracted by paired test comparisons between groups. Thyroid stimulant hormone (TSH); sample size (*n*).

**Table 3 nutrients-14-01326-t003:** Clinical outcomes between groups.

	Lean (*n* = 20)	Obesity (*n* = 20)	*p*-Value
Maternal complications			
Preeclampsia	4 (20.0%)	2 (10.0%)	0.661 ^a^
Pregnancy-induced hypertension	1 (5.0%)	5 (25.0%)	0.182 ^a^
Gestational diabetes mellitus	1 (5.0%)	4 (20.0%)	0.342 ^a^
Anemia	18 (90.0%)	15 (75.0%)	0.407 ^a^
Urinary infection	1 (5.0%)	2 (10.0%)	0.999 ^a^
Fetal complications			
Intrauterine growth restriction–Fetus 1	0 (0.0%)	2 (10.0%)	0.487 ^a^
Intrauterine growth restriction–Fetus 2	2 (10.0%)	1 (5.0%)	0.999 ^a^
Malformations–Fetus 1	3 (15.0%)	0 (0.0%)	0.231 ^a^
Malformations–Fetus 2	3 (15.0%)	2 (10.0%)	0.999 ^a^
Obstetrical complications			
Premature rupture of membrane	1 (5.0%)	4 (20.0%)	0.342 ^a^
Threat of premature labor	1 (5.0%)	2 (10.0%)	0.999 ^a^
C-section	13 (65.0%)	16 (80.0%)	0.479 ^a^
Puerperal hemorrhage	1 (5.0%)	1 (5.0%)	0.999 ^a^
Neonatal variables and complications			
Gestational age (weeks)	37.5 [36.5; 38.0]	37.1 [35.0; 37.4]	0.083 ^b^
Prematurity	6 (30.0%)	9 (45.0%)	0.514 ^a^
Birth body weight-neonate 1 (g)	2400.0 [2216.2; 2712.5]	2467.5 [1867.0; 3011.2]	0.705 ^b^
LBW-neonate 1	2 (10.0%)	6 (30.0%)	0.235 ^a^
Birth body weight-neonate 2 (g)	2602.5 [2098.8; 2783.8]	2500.0 [2141.2; 2887.5]	0.725 ^b^
LBW-neonate 2	3 (15.0%)	4 (20.0%)	0.999 ^a^
Apgar 5 min-neonate 1	10.0 [9.0; 10.0]	10.0 [9.0; 10.0]	0.890 ^b^
Apgar 5 min-neonate 2	10.0 [9.0; 10.0]	10.0 [9.0; 10.0]	0.436 ^b^
Artery pH-neonate 1	7.3 [7.3; 7.3]	7.3 [7.3; 7.3]	0.211 ^b^
Artery pH-neonate 2	7.3 [7.3; 7.3]	7.3 [7.3; 7.3]	0.507 ^b^
NICU admission-neonate 1	8 (40.0%)	8 (40.0%)	0.999 ^a^
RDS-neonate 1	1 (5.0%)	4 (20.0%)	0.342 ^a^
NICU admission-neonate 2	8 (40.0%)	7 (35.0%)	0.999 ^a^
RDS-neonate 2	3 (15.0%)	4 (20.0%)	0.999 ^a^
Neonatal death	0 (0.0%)	1 (5.0%)	0.999 ^a^

In quantitative variables, the data show median and interquartile range [Q1; Q3]. In qualitative variables, the sample size (*n*) and relative frequency (%) are shown. The *p*-Value was extracted by ^a^ Fisher’s and ^b^ Mann–Whitney tests. Low birth weight (LBW); Neonatal Intensive Care Unit (NICU); respiratory distress syndrome (RDS).

## Data Availability

The data presented in this study are available on request from the corresponding author. The availability of the data is restricted to investigators based in academic institutions.

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
