# Peer review of "Maternal Obesity in Twin Pregnancy: The Role of Nutrition to Reduce Maternal and Fetal Complications"

_nutrients, 2022, doi:10.3390/nu14071326_

Round 1

Reviewer 1 Report

Dear authors, I am pleased to review your manuscript. I agree that studying maternal nutritional status in twin pregnancies has growing importance in both developed and developing countries. Although interesting, I have some comments regarding your manuscript, and I only intend to improve it. 

  1. At first reading, your title suggests a clinical trial. Afterward, in the Results and Discussion, you mention your local protocols of follow-up. Is your paper a description of your expertise within obese twin pregnancies? If yes, I would recommend explaining this in more detail in the Methods. For example, how this nutritional follow-up was performed (number of visits, amount of calories, exercises prescription, etc.)? Why have you chosen two-time points for describing laboratory results, since you have at least four measurements? 
  1. In my opinion, the final sample needs more explanation. Did you have 223 twin pregnancies in total, or this number was only obese women? Please clarify your inclusion and exclusion criteria in the Methods. For example, why have you excluded preterm birth <36w? I would recommend using a flowchart (e.g., the STROBE model). Additionally, in Table 2, there are missing values for laboratory tests – how have you dealt with this data?  
  1. The Discussion lacks interpretation. In the Conclusion, what is the generalizability of your findings? 
  1. The English language needs extensive reviewing by an expert.  

Author Response

Dear authors, I am pleased to review your manuscript. I agree that studying maternal nutritional status in twin pregnancies has growing importance in both developed and developing countries. Although interesting, I have some comments regarding your manuscript, and I only intend to improve it.

Response: Thank you for taking the time to review our work. Kindly, see our response to your very accurate comment.

At first reading, your title suggests a clinical trial. Afterward, in the Results and Discussion, you mention your local protocols of follow-up. Is your paper a description of your expertise within obese twin pregnancies? If yes, I would recommend explaining this in more detail in the Methods. For example, how this nutritional follow-up was performed (number of visits, amount of calories, exercises prescription, etc.)? Why have you chosen two-time points for describing laboratory results, since you have at least four measurements?

Response: This is a very good appreciation, we wanted to reflect our experience as a national follow-up center in high-risk pregnancies, therefore, and following the reviewer's comment, we have expanded the protocol for nutritional follow-up of obese women with twin gestation (lines 109-117). Also, we controlled the biochemical variables at the beginning and at the end of gestation, therefore we collected the variables at these key points, this was also explained in the material and methods (lines 142-143). However, we believe it is convenient to include it in the limitations of the study as they could vary in mid-pregnancy (lines 372-373).

In my opinion, the final sample needs more explanation. Did you have 223 twin pregnancies in total, or this number was only obese women? Please clarify your inclusion and exclusion criteria in the Methods. For example, why have you excluded preterm birth <36w? I would recommend using a flowchart (e.g., the STROBE model).

Response: This is a great point. The women with twin pregnancies attended at HULP were 233. However, to avoid confounding factors, we excluded those who might have had risk factors prior to pregnancy. Also, we excluded those who had abortion at the beginning of gestation and those who we did not know their BMI at the beginning of gestation. To do more understandable, we have included flow-chart as figure 1 in the text. Pregnancies under 36w were not excluded in the study.

Additionally, in Table 2, there are missing values for laboratory tests – how have you dealt with this data?

Response: Missing data were not imputed in this study. Therefore, the description of the qualitative variables was approached by knowing the distribution (Shapiro test). Then, the hypothesis test was applied according to this distribution (Mann-Whitney or Student’s t tests), ensuring that the data that did not follow normality could be described by the median and not the mean.

The Discussion lacks interpretation. In the Conclusion, what is the generalizability of your findings?

Response: We strongly disagree with this comment. Therefore, we have implemented new information into the discussion, adding multiple paragraphs on the importance of proper screening for vitamin D and iron levels. As well as the need for supplementation in those women who could benefit to prevent complications and the importance of healthy lifestyles. We have added two new references (24 and 30) that support this evidence. In other hand, we have added a further interpretation and generalized the final conclusion.

The English language needs extensive reviewing by an expert.

Response: Thank you for this consideration, the English was reviewed by a native speaker.

Reviewer 2 Report

In this paper authors enrolled 40 twin pregnancies divided in two groups (20 lean and 20 obese). It has been found that hemoglobin levels in obese women were lower than lean women. In addition, the tendency of glucose and TSH levels, and platelets were to increase in obese compared to lean women. In the third trimester, the TSH levels were higher in obese than lean women while no significant differences were detected in the rate of maternal, fetal, obstetrical, and neonatal complications between the groups.

This would be a very interesting study but the cohort is very small. In this type of study the cohort plays a key role and this is too small to show possible association between the two groups. The proof is that obesity and overweight are known risk factors for Preeclampsia and GDM (PMID: 33549038, 30297319, 31536940, 34283479 and many others) but the authors did not find any association. 

The reader suggests to increase the sample size. 

Author Response

In this paper authors enrolled 40 twin pregnancies divided in two groups (20 lean and 20 obese). It has been found that hemoglobin levels in obese women were lower than lean women. In addition, the tendency of glucose and TSH levels, and platelets were to increase in obese compared to lean women. In the third trimester, the TSH levels were higher in obese than lean women while no significant differences were detected in the rate of maternal, fetal, obstetrical, and neonatal complications between the groups.

Response: Thank you for taking the time to review our work. Kindly, see our response to your very accurate comment.

This would be a very interesting study, but the cohort is very small. In this type of study, the cohort plays a key role and this is too small to show possible association between the two groups. The proof is that obesity and overweight are known risk factors for Preeclampsia and GDM (PMID: 33549038, 30297319, 31536940, 34283479 and many others) but the authors did not find any association.

The reader suggests to increase the sample size.

Response: We fully agree with this comment, there is a lot of evidence to support that an elevated BMI before to pregnancy is a risk factor for obstetric complications. However, there are few studies that look at twin pregnancies, and this is the relevance of our data. In addition, these women with twin pregnancy and high BMI before pregnancy were under strict follow-up by the nutrition unit of our hospital, which decreased weight gain during pregnancy, one of the determining variables in the development of these complications. Unfortunately, we were unable to increase the sample size, since twin pregnancies whose mothers have a high BMI before pregnancy have a low prevalence in our health care context.

Reviewer 3 Report

Review Report – original article Maternal Obesity in Twin Pregnancy: The Role of Nutrition to 2 Reduce Maternal and Fetal Complications

The prevalence of the twin pregnancy is increasing due to the due to the increasing use of in vitro fertilization methods and of ovulation-inducing drugs. Twin pregnancy is considered as beinng one of high-risk, associated with a lower gestational age, a higher risk of low weight for gestational age newborns and neonatal complications related to prematurity. It is well known that the need for C-section is more prevalent in twin pregnancy.  Low for gestational newborns have an increse risk for metabolic disorders and cadriovascular disease in the adult age and epigenetic mechanisms had been involved in this association. A large number of studies adressed the association between maternal undernutrition and small for gestational age babies. Obesity is consider as a pandemic problem, affecting high-income and developing countries, and the consequences of high prepregnancy BMI represent an important issue to be studied.

The studied entitled Maternal Obesity in Twin Pregnancy: The Role of Nutrition to 2 Reduce Maternal and Fetal Complications, aim to compare the biochemical and hematological parameters along gestation in single pregnancy with those in twin prgnancies. The manuscript is clear, relevant for the field and presented in a well-structured manner. The manuscript  isscientifically sound and the experimental design is  appropriate to test the hypothesis. Tables are appropriate and easy to interpret and understand.

The cited references are mostly within the last 5 years.

Minor revisions:

  1. The methodology of the study is according to the proposed hypotesis, but close nutritional follow-up (line 87-91) should be more carefully described in terms of composition or dietary pattern indicated during pregnancy in both studied groups. It should be made a clearer specification whether the group of pregnant women with obesity and multiple pregnancies benefited only from nutritional counseling or from a nutritional intervention, and also if it was different from the control group.
  2. The conclusions are consistent with the evidence and arguments presented, but since the nutritional follow up of the obese women was included in the hypoteses (line 94), it should be mention in the conclusion section, too.

Author Response

The prevalence of the twin pregnancy is increasing due to the due to the increasing use of in vitro fertilization methods and of ovulation-inducing drugs. Twin pregnancy is considered as being one of high-risk, associated with a lower gestational age, a higher risk of low weight for gestational age newborns and neonatal complications related to prematurity. It is well known that the need for C-section is more prevalent in twin pregnancy.  Low for gestational newborns have an increased risk for metabolic disorders and cardiovascular disease in the adult age and epigenetic mechanisms had been involved in this association. A large number of studies addressed the association between maternal undernutrition and small for gestational age babies. Obesity is considered as a pandemic problem, affecting high-income and developing countries, and the consequences of high prepregnancy BMI represent an important issue to be studied.

Response: We would like to thank the reviewer for the time spent on our article, as well as for the words. We fully agree with this consideration, the implication of overweight/obesity is a risk factor in cardiometabolic disease, even more in twin pregnancies.

The studied entitled Maternal Obesity in Twin Pregnancy: The Role of Nutrition to Reduce Maternal and Fetal Complications, aim to compare the biochemical and hematological parameters along gestation in single pregnancy with those in twin prgnancies. The manuscript is clear, relevant for the field and presented in a well-structured manner. The manuscript  isscientifically sound and the experimental design is  appropriate to test the hypothesis. Tables are appropriate and easy to interpret and understand.

The cited references are mostly within the last 5 years.

Response: Thank you again for these kind words. We consider twin pregnancy as a risk factor itself that has been excluded from many obstetric studies. However, twin pregnancy could have considerations. We wanted to see what happened with maternal overweight before pregnancy and to highlight the fundamental role that nutrition units play in the management of this type of pregnancy.

Minor revisions:

  • The methodology of the study is according to the proposed hypothesis, but close nutritional follow-up (line 87-91) should be more carefully described in terms of composition or dietary pattern indicated during pregnancy in both studied groups. It should be made a clearer specification whether the group of pregnant women with obesity and multiple pregnancies benefited only from nutritional counseling or from a nutritional intervention, and also if it was different from the control group.

Response: Nutritional recommendations were described in more detail in the methos section of the study (lines 109-117).

  • The conclusions are consistent with the evidence and arguments presented, but since the nutritional follow up of the obese women was included in the hypotheses (line 94), it should be mentioned in the conclusion section, too.

Response: This consideration was described in the conclusions as proposed by the reviewer (lines 368-371).

Round 2

Reviewer 2 Report

the manuscript has been improved by adding a flow chart and new information about the study cohort also explaining the sample size analysed. Now the manuscript is way more clear and can be accepted for publication in the present form.